# Reward Mapping for Transfer in Long-Lived Agents

**Xiaoxiao Guo**
Computer Science and Eng.
University of Michigan
guoxiao@umich.edu

**Satinder Singh**
Computer Science and Eng.
University of Michigan
baveja@umich.edu

**Richard Lewis**
Department of Psychology
University of Michigan
rickl@umich.edu

## Abstract

We consider how to transfer knowledge from previous tasks (MDPs) to a current task in long-lived and bounded agents that must solve a sequence of tasks over a finite lifetime. A novel aspect of our transfer approach is that we reuse *reward functions*. While this may seem counterintuitive, we build on the insight of recent work on the optimal rewards problem that guiding an agent's behavior with reward functions other than the task-specifying reward function can help overcome computational bounds of the agent. Specifically, we use good guidance reward functions learned on previous tasks in the sequence to incrementally train a *reward mapping function* that maps task-specifying reward functions into good initial guidance reward functions for subsequent tasks. We demonstrate that our approach can substantially improve the agent's performance relative to other approaches, including an approach that transfers policies.

## 1 Introduction

We consider agents that live for a long time in a sequential decision-making environment. While many different interpretations are possible for the notion of *long-lived*, here we consider agents that have to solve a sequence of tasks over a continuous lifetime. Thus, our problem is closely related to that of transfer learning in sequential decision-making, which can be thought of as a problem faced by agents that have to solve a set of tasks. Transfer learning [18] has explored the *reuse* across tasks of many different components of a reinforcement learning (RL) architecture, including value functions [16, 5, 8], policies [9, 20], and models of the environment [1, 17]. Other transfer approaches have considered parameter transfer [19], selective reuse of sample trajectories from previous tasks [7], as well as reuse of learned abstract representations such as options [12, 6].

A novel aspect of our transfer approach in long-lived agents is that we will reuse reward functions. At first blush, it may seem odd to consider using a reward function different from the one specifying the current task in the sequence (indeed, in most RL research rewards are considered an immutable part of the task description). But there is now considerable work on designing good reward functions, including reward-shaping [10], inverse RL [11], optimal rewards [13] and preference-elicitation [3]. In this work, we specifically build on the insight of the optimal rewards problem (ORP; described in more detail in the next section) that guiding an agent's behavior with reward functions other than the task-specifying reward function can help overcome computational bounds in the agent architecture. We base our work on an algorithm from Sorg et.al. [14] that learns good guidance reward functions incrementally in a single-task setting.

Our main contribution in this paper is a new approach to transfer in long-lived agents in which we use good guidance reward functions learned on previous tasks in the sequence to incrementally train a *reward mapping function* that maps task-specifying reward functions into good initial guidance reward functions for subsequent tasks. We demonstrate that our approach can substantially improve a long-lived agent's performance relative to other approaches, first on an illustrative grid world domain, and second on a networking domain from prior work [9] on the reuse of policies for transfer.

In the grid world domain only the task-specifying reward function changes with tasks, while in the networking domain both the reward function and the state transition function change with tasks.

## 2 Background: Optimal Rewards for Bounded Agents in Single Tasks

We consider sequential decision-making environments formulated as controlled Markov processes (CMPs); these are defined via a state space $S$, an action space $A$, and a transition function $T$ that determines a distribution over next states given a current state and action. A task in such a CMP is defined via a reward function $R$ that maps state-action pairs to scalar values. The objective of the agent in a task is to execute the optimal policy, i.e., to choose actions in such a way as to optimize *utility* defined as the expected value of cumulative reward over some lifetime. A CMP and reward function together define a Markov decision process or MDP; hence tasks in this paper are MDPs.

There are many approaches to planning an optimal policy in MDPs. Here we will use UCT [4] which incrementally plans the action to take in the current state. It simulates a number of trajectories from the current state up to some maximum depth, choosing actions at each point based on the sum of an estimated action-value that encourages exploitation and a reward bonus that encourages exploration. It has theoretical guarantees of convergence and works well in practice on a variety of large-scale planning problems. We use UCT in this paper because it is one of the state of the art algorithms in RL planning and because there exists a good optimal reward finding algorithm for it [14].

**Optimal Rewards Problem (ORP).**  In almost all of RL research, the reward function is considered part of the task specification and thus unchangeable. The optimal reward framework of Singh et al. [13] stems from the observation that a reward function plays two roles simultaneously in RL problems. The first role is that of *evaluation* in that the task-specifying reward function is used by the agent designer to evaluate the actual behavior of the agent. The second is that of *guidance* in that the reward function is also used by the RL algorithm implemented by the agent to determine its behavior (e.g., via Q-learning [21] or UCT planning [4]). The optimal rewards problem separates these two roles into two separate reward functions, the task-specifying *objective* reward function used to evaluate performance, and an *internal* reward function used to guide agent behavior. Given a CMP $M$, an objective reward function $R^o$, an agent $\mathcal{A}$ parameterized by an internal reward function, and a space of possible internal reward functions $\mathcal{R}$, an optimal internal reward function $R^{i^*}$ is defined as follows (throughout superscript $o$ will denoted objective evaluation quantities and superscript $i$ will denote internal quantities):

$$R^{i^*} = arg \max_{R^i \in \mathcal{R}} \mathbb{E}_{h \sim \langle \mathcal{A}(R^i), M \rangle} \Big\{ U^o(h) \Big\},$$

where $\mathcal{A}(R^i)$ is the agent with internal reward function $R^i$, $h \sim \langle \mathcal{A}(R^i), M \rangle$ is a random history (trajectory of alternating states and actions) obtained by the interaction of agent $\mathcal{A}(R^i)$ with CMP $M$, and $U^o(h)$ is the objective utility (as specified by $R^o$) to the agent designer of interaction history $h$. The optimal internal reward function will depend on the agent $\mathcal{A}$'s architecture and its limitations, and this distinguishes ORP from other reward-design approaches such as inverse-RL. When would the optimal internal reward function be different from the objective reward function? If an agent is *unbounded* in its capabilities with respect to the CMP then the objective reward function is always an optimal internal reward function. More crucially though, in the realistic setting of *bounded* agents, optimal internal reward functions may be quite different from objective reward functions. Singh et al.[13] and Sorg et al.[14] provide many examples and some theory of when a good choice of internal reward can mitigate agent bounds, including bounds corresponding to limited lifetime to learn [13], limited memory [14], and limited resources for planning (the specific bound of interest in this paper).

**PGRD: Solving the ORP on-line while planning.**  Computing $R^{i^*}$ can be computationally non-trivial. We will use Sorg et.al.'s [14, 15] policy gradient reward design (PGRD) method that is based on the insight that any planning algorithm can be viewed as procedurally translating the internal reward function $R^i$ into behavior—that is, $R^i$ are indirect parameters of the agent's policy. PGRD cheaply computes the gradient of the objective utility with respect to the $R^i$ parameters through UCT planning. Specifically, it takes a simulation model of the CMP and an objective reward function and uses UCT to simultaneously plan actions with respect to the current internal reward function as well as to update the internal reward function in the direction of the gradient of the objective utility for use in the next planning step.

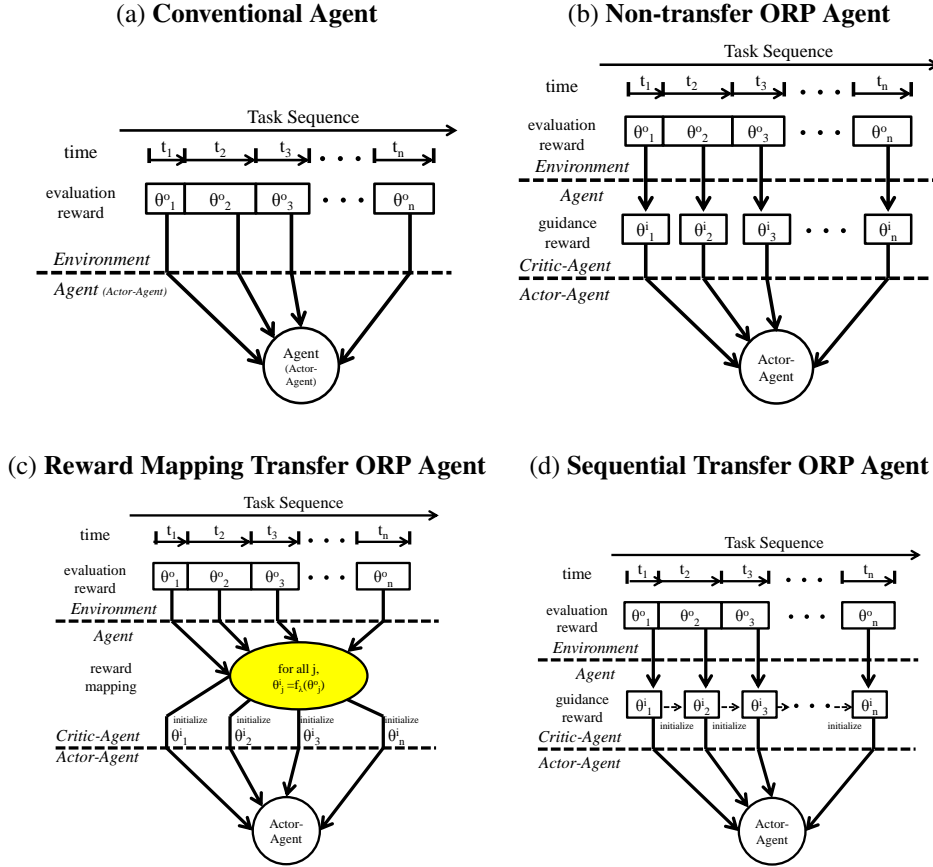

Figure 1: The four agent types compared in this paper. In each figure, time flows from left to right. The sequence of objective reward parameters and task durations for $n$ tasks are shown in the environment portion of each figure. In figures (b-d) the agent portion of the figure is further split into a critic-agent and an actor-agent; figure (a) does not have this split because it is the conventional agent. The critic-agent translates the objective reward parameters $\theta^o$ into the internal reward parameters $\theta^i$. The actor-agent is a UCT agent in all our implementations. The critic-agent component varies across the figures and is crucial to understanding the differences among the agents (see text for detailed descriptions).

## 3 Four Agent Architectures for the Long-Lived Agent Problem

**Long-Lived Agent's Objective Utility.** We will consider the case where objective rewards are linear functions of objective reward features. Formally, the $j^{th}$ task is defined by objective reward function $R_j^o(s, a) = \theta_j^o \cdot \psi^o(s, a)$, where $\theta_j^o$ is the parameter vector for the $j^{th}$ task, $\psi^o$ are the task-independent objective reward features of state and action, and '$\cdot$' denotes the inner-product. Note that the features are constant across tasks while the parameters vary. The $j^{th}$ task lasts for $t_j$ time steps. Given some agent $\mathcal{A}$ the expected objective utility achieved for a particular task sequence $\{\theta_j^o, t_j\}_{j=1}^K$, is $\mathbb{E}_{h \sim \langle \mathcal{A}, M \rangle} \sum_{j=1}^K \left\{ U^{\theta_j^o}(h_j) \right\}$, where for ease of exposition we denote the history during task $j$ simply as $h_j$. In general, there may be a distribution over task sequences, and the expected objective utility would then be a further expectation over such a distribution.

In some transfer or other long-lived agent research, the emphasis is on *learning* in that the agent is assumed to lack complete knowledge of the CMP and the task specifications. Our emphasis here is on *planning* in that the agent is assumed to know the CMP perfectly as well as the task specifications as they change. If the agent were unbounded in planning capacity, there would be nothing interesting left to consider because the agent could simply find the optimal policy for each new task and execute it. What makes our problem interesting therefore is that our UCT-based planning agent is computationally limited: the depth and number of trajectories feasible are small enough (relative

to the size of the CMP) that it cannot find near-optimal actions. This sets up the potential for both the use of the ORP and of transfer across tasks. Note that basic UCT does use a reward function but does not use an initial value function or policy and hence changing a reward function is a natural and consequential way to influence UCT. While non-trivial modifications of UCT could allow use of value functions and/or policies, we do not consider them here. In addition, in our setting a model of the CMP is available to the agent and so there is no scope for transfer by reuse of model knowledge. Thus, our reuse of reward functions may well be the most consequential option available in UCT.

Next we discuss four different agent architectures represented graphically in Figure 1, starting with a conventional agent that ignores both the potential of transfer and that of ORP, followed by three different agents that do not to varying degrees.

**Conventional Agent.** Figure 1(a) shows the baseline conventional UCT-based agent that ignores the possibility of transfer and treats each task separately. It also ignores ORP and treats each task's objective reward as the internal reward for UCT planning during that task.

The remaining three agents will all consider the ORP, and share the following details: The space of internal reward functions $\mathcal{R}$ is the space of all linear functions of internal reward features $\psi^i(s, a)$, i.e., $\mathcal{R}(s, a) = \{\theta \cdot \psi^i(s, a)\}_{\theta \in \Theta}$, where $\Theta$ is the space of possible parameters $\theta$ (in this paper all finite vectors). Note that the internal reward features $\psi^i$ and the objective reward features $\psi^o$ do not have to be identical.

**Non-Transfer ORP Agent.** Figure 1(b) shows the non-transfer agent that ignores the possibility of transfer but exploits ORP. It initializes the internal reward function to the objective reward function of each new task as it starts and then uses PGRD to adapt the internal reward function while acting in that task. Nothing is transferred across task boundaries. This agent was designed to help separate the contributions of ORP and transfer to performance gains.

**Reward-Mapping-Transfer ORP Agent.** Figure 1(c) shows the reward-mapping agent that incorporates our main new idea. It exploits both transfer and ORP via incrementally learning a reward mapping function. A reward mapping function $f$ maps objective reward function parameters to internal reward function parameters: $\forall j, \theta_j^i = f(\theta_j^o)$. The reward mapping function is used to initialize the internal reward function at the beginning of each new task. PGRD is used to continually adapt the initialized internal reward function throughout each task.

The reward mapping function is incrementally trained as follows: when task $j$ ends, the objective reward function parameters $\theta_j^o$ and the adapted internal reward function parameters $\hat{\theta}_j^i$ are used as an input-output pair to update the reward mapping function. In our work, we use nonparametric kernel-regression to learn the reward mapping function. Pseudocode for a general reward mapping agent is presented in Algorithm 1.

**Sequential-Transfer ORP Agent.** Figure 1(d) shows the sequential-transfer agent. It also exploits both transfer and ORP. However, it does not use a reward mapping function but instead continually updates the internal reward function across task boundaries using PGRD. The internal reward function at the end of a task becomes the initial internal reward function at the start of the next task achieving a simple form of sequential transfer.

## 4 Empirical Evaluation

The four agent architectures are compared to demonstrate that the reward mapping approach can substantially improve the bounded agent's performance, first on an illustrative grid world domain, and second on a networking routing domain from prior work [9] on the transfer of policies.

### 4.1 Food-and-Shelter Domain

The purpose of the experiments in this domain are (1) to systematically explore the relative benefits of the use of ORP, and of transfer (with and without the use of the reward-mapping function), each in isolation and together, (2) to explore the sensitivity and dependence of these relative benefits on parameters of the long-lived setting such as mean duration of tasks, and (3) to visualize what is learned by the reward mapping function.

---

**Algorithm 1** General pseudocode for Reward Mapping Agent (Figure 1(c))

---

1: Input: $\{\theta^o_j, t_j\}^k_{j=1}$, where $j$ is task indicator, $t_j$ is task duration, and $\theta^o_j$ are the objective reward function parameters specifying task $j$.

2:
3: **for** $t = 1, 2, 3, ...$ **do**
4:     **if** a new task $j$ starts **then**
5:         obtain current objective reward parameters $\theta^o_j$
6:         compute: $\theta^i_j = f(\theta^o_j)$
7:         initialize the internal reward function using $\theta^i_j$
8:     **end if**
9:     $a_t :=$ planning$(s_t; \theta^i_j)$   (select action using UCT guided by reward function $\theta^i_j$)
10:    $(s_{t+1}, r_{t+1}) :=$ takeAction$(s_t, a_t)$
11:    $\theta^i :=$ updateInternalRewardFunction$(\theta^i, s_t, a_t, s_{t+1}, r_{t+1})$   (via PGRD)
12:
13:    **if** current task ends **then**
14:        obtain current internal reward parameters as $\hat{\theta}^i_j$
15:        update reward mapping function $f$ using training pair $(\theta^o, \hat{\theta}^i_j)$
16:    **end if**
17: **end for**

---

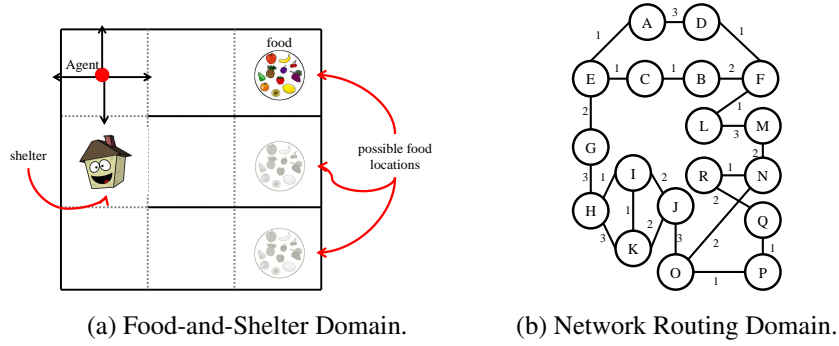

(a) Food-and-Shelter Domain.          (b) Network Routing Domain.

Figure 2: Domains used in empirical evaluation; the network routing domain comes from [9].

The environment is a simple 3 by 3 maze with three left-to-right corridors. Thick black lines indicate impassable walls. The position of the shelter and possible positions of food are shown in Figure 2.

*Dynamics.* The shelter breaks down with a probability of 0.1 at each time step. Once the shelter is broken, it remains broken until repaired by the agent. Food appears at the rightmost column of one of the three corridors and can be eaten by the agent when the agent is at the same location with the food. When food is eaten, new food reappears in a different corridor. The agent can move in four cardinal directions, and every movement action has a probability of 0.1 to result in movement in a random direction; if the direction is blocked by a wall or the boundary, the action results in no movement. The agent eats food and repairs shelter automatically whenever collocated with food and shelter respectively. The discount factor $\gamma = 0.95$.

*State.* A state is a tuple $(l, f, h)$, where $l$ is the location of the agent, $f$ is the location of the food, and $h$ indicates whether the shelter is broken.

*Objective Reward Function.* At each time step, the agent receives a positive reward of $e$ (the eat-bonus) for eating food and a negative reward of $b$ (the broken-cost) if the shelter is broken. Thus, the objective reward function's parameters are $\theta^o_j = (e_j, b_j)$, where $e_j \in [0, 1]$ and $b_j \in [-1, 0]$. Different tasks will require the agent to behave in different ways. For example, if $(e_j, b_j) = (1,0)$, the agent should explore the maze to eat more food. If $(e_j, b_j) = (0, -1)$, the agent should remain at the shelter's location in order to repair the shelter as it breaks.

*Space of Internal Reward Functions.* The internal reward function is $R^i_j(s) = R^o_j(s) + \theta^i_j\psi^i(s)$, where $R^o_j(s)$ is the objective reward function, $\psi^i(s) = 1 - \frac{1}{n_l(s)}$ is the inverse recency feature

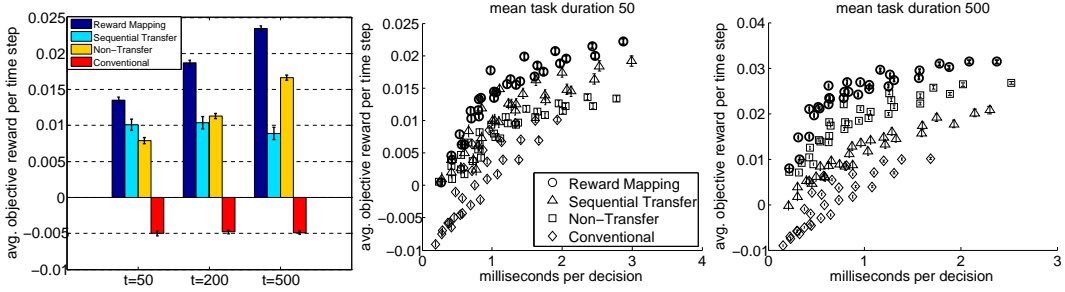

Figure 3: *(Left)* Performance of four agents in food-and-shelter domain at three different mean task durations. *(Middle and Right)* Comparing performance while accounting for computational overhead of learning and using the reward mapping function. See text for details.

and $n_l(s)$ is the number of time steps since the agent's last visit to the location in state $s$. Since there is exactly one internal reward parameter, $\theta_j^i$ is a scalar. A positive $\theta_j^i$ encourages the agent to visit locations not visited recently, and a negative $\theta_j^i$ encourages the agent to visit locations visited recently.

**Results: Performance advantage of reward mapping.** 100 sequences of 200 tasks were generated, with Poisson distributions for task durations, and with objective reward function parameters sampled uniformly from their ranges. The agents used UCT with depth 2 and 500 trajectories; the conventional agent is thereby bounded as evidenced in its poor performance (see Figure 3).

The left panel in Figure 3 shows average objective reward per time step (with standard error bars). There are three sets of four bars each where each bar within a set is for a different architecture (see legend), and each set is for a different mean task duration (50, 200, and 500 from left to right). For each task duration the reward mapping agent does best and the conventional agent does the worst. These results demonstrate transfer helps performance and that transfer via the new reward mapping approach can substantially improve a bounded long-lived agent's performance relative to transfer via the competing method of sequential transfer. As task durations get longer the ratio of the reward-mapping agent's performance to the non-transfer agent's performance get smaller, though remains > 1 (by visually taking the ratio of the corresponding bars). This is expected because the longer the task duration the more time PGRD has to adapt to the task, and thus the less the better initialization provided by the reward mapping function matters.

In addition, the sequential transfer agent does better than the non-transfer agent for the shortest task duration of 50 while the situation reverses for the longest task duration of 500. This is intuitive and significant as follows. Recall that the initialization of the internal reward function from the final internal reward function of the previous task can hurt performance in the sequential transfer setting if the current task requires quite different behavior from the previous—but it can help if two successive tasks are similar. Correcting the internal reward function

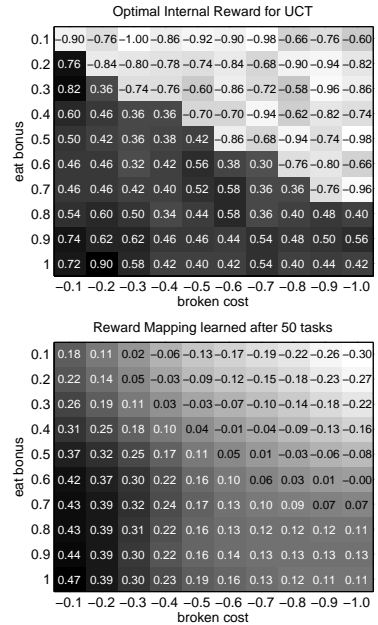

Figure 4: Reward mapping function visualization: *Top:* Optimal mapping, *Bottom:* Mapping found by the Reward Mapping agent after 50 tasks.

could cost a large number of steps. These effects are exacerbated by longer task durations because the agent then has longer to adapt its internal reward function to each task. In general, as task duration increases, the non-transfer agent improves but the sequential transfer agent worsens.

**Results: Performance Comparison considering computational overhead.** The above results ignore the computational overhead incurred by learning and using the reward mapping function. The two rightmost plots in the bottom row of Figure 3 show the average objective reward per time step as a function of milliseconds per decision for the four agent architectures for a range of depth $\{1, \ldots, 6\}$, and trajectory-count $\{200, 300, \ldots, 600\}$ parameters for UCT. The plots show that for

the entire range of time-per-decision, the best performing agents are reward-mapping agents—in other words, it is *not* better to spend the overhead time of the reward-mapping on additional UCT search. This can be seen by observing that the highest dot at any vertical column on the x-axis belongs to the reward mapping agent. Thus, the overhead of the reward mapping function in the reward mapping agent is insignificant relative to the computational cost of UCT (this last cost is all the conventional agent incurs).

**Results: Reward mapping visualization.** Using a fixed set of tasks (as described above) with mean duration of 500, we estimated the optimal internal reward parameter (the coefficient of the inverse-recency feature) for UCT by a brute-force grid search. The optimal internal reward parameter is visualized as a function of the two parameters of the objective reward function (broken cost and eat bonus) in Figure 4, top. Negative coefficients (light color squares) for inverse-recency feature discourage exploration while positive coefficients (dark color squares) encourage exploration. As would be expected the top right corner (high penalty for broken shelter and low reward for eating) discourages exploration while the bottom left corner (high reward for eating and low cost for broken shelter) encourages exploration. Figure 4, bottom, visualizes the learned reward mapping function after training on 50 tasks. There is a clearly similar pattern to the optimal mapping in the upper graph, though it has not captured the finer details.

## 4.2 Network Routing Domain

The purposes of the following experiments are to (1) compare performance of our agents to a competing policy transfer method [9] from a closely related setting on a networking application domain defined by the competing method; (2) demonstrate that our reward mapping and other agents can be extended to a multi-agent setting as required by this domain; and (3) demonstrate that the reward-mapping approach can be extended to handle task changes that involve changes to the transition function as well as objective reward.

The network routing domain [9] (see Figure 2(b)) is defined from the following components. (1) A set of *routers*, or nodes. Every router has a queue to store packets. In our experiments, all queues are of size three. (2) A set of *links* between two routers. All links are bidirectional and full-duplex, and every link has a weight (uniformly sampled from $\{1,2,3\}$) to indicate the cost of transmitting a packet. (3) A set of active *packets*. Every packet is a tuple *(source, destination, alive-time)*, where *source* is the node which generated the packet, *destination* is the node that the packet is sent to, and *alive-time* is the time period that the packet has existed in the network. When a packet is delivered to its destination node, the alive-time is the end-to-end delay. (4) A set of *packet generators*. Every node has a packet generator that specifies a stochastic method to generate packets. (5) A set of *power consumption functions*. Every node's power consumption at time $t$ is the number of packets in its queue multiplied by a scalar parameter sampled uniformly in the range $[0, 0.5]$.

*Actions, dynamics, and states.* Every node makes its routing decision separately and has its own action space (these determine which neighbor the first packet in the queue is sent to). If multiple packets reach the same node simultaneously, they are inserted into the queue in random order. Packets that arrives after the queue is full cause network congestion and result in packet loss. The global state at time $t$ consists of the contents of all queues at all nodes at $t$.

*Transition function.* In a departure from the original definition of the routing domain, we parameterize the transition function to allow a comparison of agents' performance when transition functions change. Originally, the state transition function in the routing problem was determined by the fixed network topology and by the parameters of the packet generators that determined among other things the destination of packets. In our modification, nodes in the network are partitioned into three groups ($G_1$, $G_2$, and $G_3$) and the probabilities that the destination of a packet belongs to each group of nodes ($p^{G_1}$, $p^{G_2}$, and $p^{G_3}$) are parameters we manipulate to change the state transition function.

*Objective reward function.* The objective reward function is a linear combination of three objective reward features, the *delay* measured as the sum of the *inverse* end-to-end delay of all packets received at all nodes at time $t$, the *loss* measured as the number of lost packets at time $t$, and *power* measured as the sum of the power consumption of all nodes at time $t$. The weights of these three features are the parameters of the objective reward function. The weight for the delay feature $\in (0, 1)$, while the weights for both loss and power are $\in (-0.2, 0)$; different choices of these weights correspond to different objective reward functions.

*Internal reward function.* The internal reward function for the agent at node $k$ is $R^i_{j,k}(s,a) = R^o_j(s,a) + \theta^i_{j,k}\psi^i_k(s,a)$, where $R^o_j(s,a)$ is the objective reward function, $\psi^i_k(s,a)$ is a binary feature vector with one binary feature for each (packet destination, action) pair. It sets the bits corresponding to the destination of the first packet in node k's queue at state $s$ and action $a$ to 1; all other bits are set to 0. The internal reward features are capable of representing arbitrary policies (and thus we also implemented classical policy gradient with these features using OLPOMDP [2] but found it to be far slower than the use of PGRD with UCT and hence don't present those results here).

**Extension of Reward Mapping Agent to handle transition function changes.** The parameters describing the transition function are concatenated with the parameters defining the objective reward function and used as input to the reward mapping function (whose output remains the initial internal reward function).

**Handling Multi-Agency.** Every nodes' agent observes the full state of the environment. All agents make decisions independently at each time step. Nodes do not know other nodes' policies, but can observe how the other nodes have acted in the past and use the empirical counts of past actions to sample other nodes' actions accordingly during UCT planning.

**Competing policy transfer method.** The competing policy transfer agent from [9] reuses policy knowledge across tasks based on a model-based average-reward RL algorithm. Their method keeps a library of policies derived from previous tasks and for each new task chooses an appropriate policy from the library and then improves the initial policy with experience. Their policy selection criterion was designed for the case when only the linear reward parameters change. However, in our experiments, tasks could differ in three different ways: (1) only reward functions change, (2) only transition functions change, and (3) both reward functions and transition functions change. Their policy selection criterion is applied to cases (1) and (3). For case (2), when only transition functions change, their method is modified to select the library-policy whose transition function parameters are closest to the new transition function parameters.

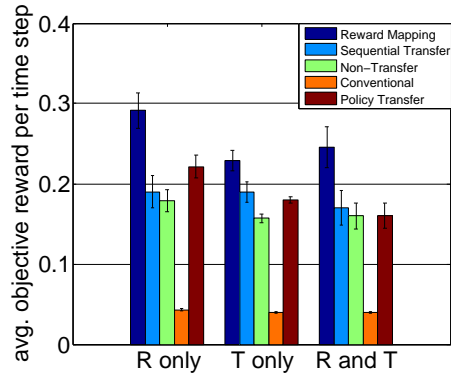

Figure 5: Performance on the network routing domain. (Left) tasks differ in objective reward functions (R) only. (Middle) tasks differ in transition function (T) only. (Right) tasks differ in both objective reward and transition (R and T) functions. See text for details.

**Results: Performance advantage of Reward Mapping Agent.** Three sets of 100 task sequences were generated, one in which the tasks differed in objective reward function only, another in which they differed in state transition function only, and third in which they differed in both. Figure 5 compares the average objective reward per time step for all four agents defined above as well as the competing policy transfer agent on the three sets. In all cases, the reward-mapping agent works best and the conventional agent worst. The competing policy transfer agent is second best when only the reward-function changes—just the setting for which it was designed.

## 5 Conclusion and Discussion

Reward functions are a particularly consequential locus for knowledge transfer; reward functions specify what the agent is to do but not how, and can thus transfer across changes in the environment dynamics (transition function) unlike previously explored loci for knowledge transfer such as value functions or policies or models. Building on work on the optimal reward problem for single task settings, our main algorithmic contribution for our long-lived agent setting is to take good guidance reward functions found for previous objective rewards and learn a mapping used to effectively initialize the guidance reward function for subsequent tasks. We demonstrated that our reward mapping approach can outperform alternate approaches; current and future work is focused on greater theoretical understanding of the general conditions under which this is true.

**Acknowledgments.** This work was supported by NSF grant IIS-1148668. Any opinions, findings, conclusions, or recommendations expressed here are those of the authors and do not necessarily reflect the views of the sponsors.

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
