[Reviews · NeurIPS 2013]

Submitted by Assigned_Reviewer_4

This paper presents a model for transfer where synthetic reward functions are learned from previous tasks, and are mapped to new tasks to produce synthetic reward functions which speed up learning. This application of the relatively recent idea of a synthetic reward function that improves the performance of an underlying "utility" reward function to the transfer scenario is new and interesting, and the experiments are well thought out, thorough and reasonably convincing.

The major criticism I had of the paper is that it does not compare to other methods of transfer. The authors are incorrect that policy transfer is not possible in their setting; that is why sub-policies, in the form of options, are often transferred instead of entire policies. (Speaking of which, the authors should probably cite Lisa Torrey's work on policy transfer.) In this case though it's OK to skip that comparison because you could use either, both, or neither types of transfer in combination, so the comparison adds information but is not critical.

However, the failure to compare against a similar reward shaping scheme is more problematic. These two methods are effectively solving the same problem, except that reward shaping does not change the ultimate solution and a synthetic reward function does. In my opinion, these two methods (even outside of the transfer scenario) have not been adequately compared, which is odd because its trivial to learn a shaping function (it's just a value function initialization, so you're just learning a value function). This lack of comparison leaves me with significant doubts about the whole synthetic reward function enterprise generally. So I think a comparison here - where the shaping function is mapped in the same way that the reward function is - would significantly improve the paper. But perhaps that is too much to ask for in a single paper, especially since the mapped shaping function could be considered a new (though somewhat obvious) method.

The paper is very well written and was generally a pleasure to read, though this was spoiled somewhat by the repeated use of parenthetical citations as nouns. The references were poorly formatted in some cases (capitalization on "mdps", etc.)
Summary: This paper describes a novel method for transfer in reinforcement learning domains, and is well written and well executed. A better experimental comparison to reward shaping methods would have been good, but isn't totally necessary.

Submitted by Assigned_Reviewer_8

The authors consider an agent that will experience a series of sequential decision-making tasks over its lifetime in the same environment (or a similar one). They propose a method for transfering knowledge acquired in previous problems to the current problem. Specifically, they consider transfering knowledge acquired in learning optimal reward functions. The authors demonstrate the utility of their approach in two examples.

The paper builds primarily on two pieces of earlier work. The first is the optimal rewards formulation in Singh et al. (2010), where the authors define an internal reward function as one that maximizes external reward (this internal reward function may differ from the external reward function if the agent is bounded in its capabilities, for instance if it has a small planning horizon). The second is the algorithm by Sorg et al. (2010) for incremental learning of such a reward function in a single task setting.

The contribution of the current paper is to place this earlier work in a multi-task setting. As in Sorg et al (2010), the agent learns an internal reward function in each task. In addition, the agent learns a mapping from the external reward functions of the tasks it has experienced in the past to the internal reward functions it has learned by the end of those tasks. The agent uses this mapping to initialize its internal reward function at the beginning of each task.

The examples in the paper are small but sufficient to demonstrate the potential utility of the approach. In practice, the success of the algorithm will depend on the availability of good features for the mapping from external rewards to internal rewards.

The paper is well written and easy to follow. The empirical evaluation is well done, informative and useful. Figure 4, in particular, is helpful in concretely showing what the algorithm has done. Including a similar figure (or description) for the network example would be a useful addition to the paper.
Summary: The paper is not particularly innovative but it is a well-executed, useful addition to the literature on transfer learning.

Submitted by Assigned_Reviewer_9

The paper describes a method to "conserve" some earlier experiences
in solving a RL problem to later runs using different reward
functions. Importantly, the system assumes a limited power for
learning, which makes it advantageous to use a surrogate reward
function that allows the agent to learn the concrete reward. The
idea is to provide the opportunity to learn an inner reward function
as a function of the external reward function.

The idea is interesting and attractive, and the prospect of a
"bounded rationality"-type assumption behind the algorithm (although
the authors studiously - and wisely - avoid using it) renders the
method a welcome approach to a more practical (and plausible)
perspective on reinforcement learning in general scenarios.

Generally well readable, the reviewer found that the paper lost
clarity in the network routing domain. I'll mention some of the
issues in the details below.

In terms of methodology, the paper falls into the general category
of the "reward-shaping" methodology, the success of the methodology
in the examples is convincing, the general method class is, of
course, already, if not maturing, but consolidating.

- line 323: what are "trajectory-count" parameters for UCT? Number
of sample runs?

- line 332: it seems that either colours, coefficient signs or
semantics of the coefficent are inconsistent here. The text says:
"negative/dark/discouraging exploration", but that does not fit
with figure 4.

- line 370: I do not understand the point of the decomposition into
G_1, G_2, G_3? What's the purpose of it?

- line 403: I do not understand how the transition function is
modeled. Don't you use reward mapping anymore here? If you use it,
*and* you modify the transition function, how does that happen?
Please reformulate this section, it is completely unclear to me.

- line 418: What is the "competing policy transfer agent"? What
model does it use?
Summary: An interesting method for transfer learning under limited
resources. Settled in existing "reward shaping" methodology
territory, the method itself looks sufficiently original and
effective to warrant publication. Some (minor) weaknesses in the
description of the second example.
Author Feedback

Author rebuttal: We thank the reviewers for helpful reviews and given a chance will incorporate into the final version their specific suggestions on how to improve clarity and readability and we will add text to discuss relationship to provided references.

Reviewer 1: a) In the network routing domain we did compare our method against the specific policy transfer method developed by the creators of the domain. We will sharpen the writing to make this clear. Furthermore, we will attempt to describe the competing method better (though space limitations make this a challenge). b) Agreed with the reviewer that our statement that “there is no scope for transfer by reuse of value function or policies” needs revision. We will be more precise by stating that without modifications UCT does use a reward function but does *not* use an initial value function or policy and hence changing a reward function is a natural and consequential way to influence unmodified UCT. However, we do agree that non-trivial modifications of UCT could allow use of value functions and/or policies. c) We will cite and describe how Torrey etal's work on policy transfer is different from ours (thanks for a useful pointer). In their most related work on transfer with model-based planning agents their focus is on discovering for which state-action pairs their old knowledge (value-functions, reward-functions and transitions) is trustworthy. Trustworthy knowledge from old tasks is reused in the new task. Trust comes from whether their old predictions remain true in the new task. So their particular research emphasis is quite different from ours, though their overall objective of transfer is similar. Their less related work focuses on transfer with model-free RL agents using learned relational knowledge. Finally, as noted above we did compare against a policy transfer method proposed by the creators of the network routing domain. d) While in this paper our focus was on adapting reward functions to achieve transfer in computationally bounded agents and for this we used the existing optimal-rewards framework, we agree with the reviewer that the optimal-rewards-based work could use a more careful direct comparison to reward-shaping; this could be relevant future work for the authors. We note that reference Sorg etal ([15]) does show that in lookahead tree planning agents potential based reward-shaping is a special case of optimal rewards and performs worse empirically. e) We will fix the use of parenthetical citations as nouns.

Reviewer 2: Yes, availability of good features for the reward-mapping function is key both for the existence of good internal reward functions as well as for the ease of the supervised learning task that maps objective rewards to internal rewards. We will acknowledge this explicitly in a revision. Thank you.

Reviewer 3: Thank you for helpful suggestions of how to improve the writing in the network routing domain. To answer some of your questions briefly using line numbers. (323) Yes, trajectory-counts are UCT sample runs or trajectories to a prescribed depth. (332) Embarrassingly we had colors on both the numbers (these were the relevant colors) and on the grid squares (these were meant to help visualize) that were chosen in a contradictory manner (negative numbers used a dark-line font but the associated grid squares were light colored). In the cited statement we mean that negative numbers written in a dark-line font discouraged exploration. We will fix this in our graphic. (370) We will clarify how the G_1, G_2, G_3 decomposition we introduced was used to allow for intuitively interesting changes to the state-transition function across tasks. Specifically, part of the state description is the destination of packets in the system. The probabilities that the destination of a packet belonged to each of the subgraphs (p^G_1, p^G_2) became parameters we could manipulate to change the state-transition probabilities. (403) In the experiments where transition functions changed with tasks, we still used a mapping function whose *output* was the initial internal reward function, but whose *inputs* now included parameters of the state transition function (specifically p^G_1 and p^G_2 as mentioned above). (418) The competing policy transfer agent is from the reference Natarajan and Tadepalli [9] in the paper; they introduced the network routing domain to evaluate their policy transfer algorithm and we use both their domain and their algorithm as a comparison. Their method stores policies and for each new task chooses an appropriate policy from the stored ones and them improves the initial policy with experience. (They use vector-based average reward RL algorithms where each component of the vector corresponds to an objective reward function feature, and store and reuse these vectors as implicit representations of policies.)